

# Wind speed time series synthesis using a parametrized power spectral density function

Ram C. Poudel[1], Dave Corbus[2], Ian Baring-Gould [2]

[1]Appalachian State University, Boone, North Carolina 28607, USA
[2]National Renewable Energy Laboratory, Golden, Colorado 80401, USA

*Correspondence to*: Ram C. Poudel (poudelrc@appstate.edu)

**Abstract**. We propose a new method to synthesize 1 Hz wind speed and wind power time series data from the industry standard 10-minute wind turbine performance data. The method is based on a parameterized power spectral density (PSD) function decomposed into trend and random components. We illustrate the intra-timestep data synthesis
utilizing 1 Hz data from two distributed wind turbines.

## 1   Introduction

The wind energy industry adopted the standard 10-minute aggregation of resource and operational data sampled at 1 Hz following the seminal work of Hoven (Lopez-Villalobosa et al., 2021; Hoven, 1957). The 10-minute aggregation practice has been a norm of the industry for a reason. This aggregation practice is
optimal for describing the relationship between the wind speed and the power output of a wind turbine (DO & BERTHAUT-GERENTES, 2018). Many engineering tools and financial models rely on 10 minutes (IEC, 2017) or hourly data. However, such an industry-standard practice comes with significant information loss (Beretta et al., 2021). The 10-minute resource data is inadequate for sizing and dispatch of integrated storage (Poudel et al., 2021), and the power system simulation study requires data at a resolution of 1 second
or better. High-resolution (one-second) input data is critical for various integration studies (Quiroz & Reno, 2012) but such a dataset is scarce for many operational wind turbines and wind farms. The legacy 10-minute averaged operational data are not quite enough to answer site-specific design questions that involve smaller time scales interaction such as power system simulation must deal with under high renewable energy contribution scenarios.

Wind energy assessment and wind farm design practice rely both on historic and forecasted data. Because of the inherent uncertainty in the forecast, the industry still relies heavily on the historically measured dataset to estimate the long-term wind speed, annual energy production (AEP), and P50, P90, P95 values of AEP, etc. Historical data sets are generally in order of hourly time scales. Such datasets work quite well in estimating AEP or capacity factor due to the spectral gap between 6 hr to 1 minute (Lopez-Villalobosa
et al., 2021). Such an hourly resolution dataset is not good enough to characterize variability in wind power P(t) that distributed energy resources (DER) must comply with for grid integration (IEEE, 2018).

The wind energy industry could take advantage of the high-resolution data acquisition capabilities many data loggers offer nowadays. In the past, data measurement efforts were limited by tools and computation capabilities (speed and storage). Popular data analysis tools such as spreadsheets were limited by the
number of rows and columns[1], and data aggregation, slicing and dicing of data, and mining capabilities were limited to some custom software and applications. Industry-standard tools have now evolved to

---

[1] [Excel 2003: 65,536 rows and 256 columns]



converge on methods as a result of concerted efforts the industry made to standardize the energy assessment process and quantify uncertainty.

To make the legacy 10 minutes dataset work for the wider site-specific design questions, we need to develop better data synthesis capabilities based on information/statistics unique to the data in question. Such capabilities might complement the industry-standard practice. Here, the data in question is the time series of wind speed, and the unique characteristic is the energy spectra of turbulence. In recent study has demonstrated wind resource power may be possessing unique one-dimensional power spectral density (PSD), and kinetic energy spectra (of horizontal velocity). The PSD seems to be a robust description that

varies a little with the spatial aggregation of wind turbines. We utilize this unique characteristic signature of PSD to synthesize data at a 1-second resolution. This study might also inform what additional parameters of data sampling and aggregation of wind resources and wind power might complement the industry-standard 10-minute aggregation practice.

## 2   Problem Description

Let $u(t, x)$ be the instantaneous value of wind speed along the horizontal x direction. Following Reynolds' decomposition, we may write $u(t, x) = \underline{u}(t) + u'(t, x)$, where, $\underline{u}$ is an average over a time t, and $\underline{u}' = 0$. The wind flow field is often inhomogeneous hence Reynolds average has been formulated in terms of time averages. Industry-standard practice reports the average ($\underline{u}$) of wind speed and wind power $P(u)$ time series at $t = T = 10$ minutes along with the standard deviation ($\sigma_u$), max(u), and min(u). These four statistics, one

measure of central tendency and three measures of dispersion, may be called $4S = [\underline{u}, \sigma u, u(max), u(min)]$.

Figure 1 depicts some representative cases of variations of u', [P(u)]', the time of max(u), and min(u) based on data collected at 400 Hz for the CART3 (Fingersh & Johnson, 2002; Anderson et al., 2022) research wind turbine at the National Renewable Energy Laboratory (NREL) Flatiron Campus. The lower subplot depicts the inter-timestep probability density function (PDF) and cumulative density function (CDF) for

the period (T=10 minutes) of aggregation.

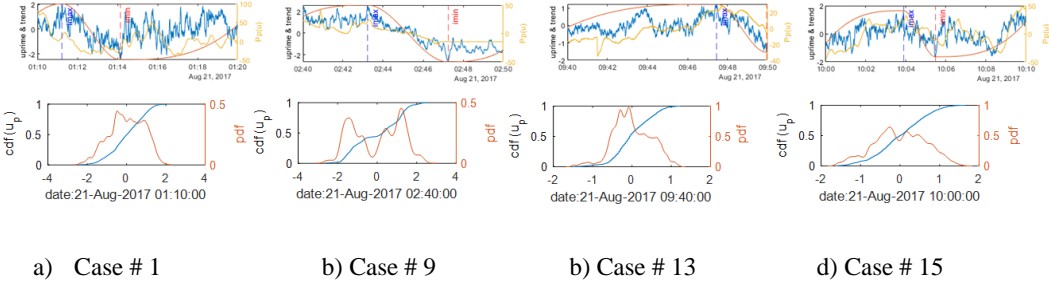

a) Case # 1      b) Case # 9      b) Case # 13      d) Case # 15

**Figure 1: Intra-timestep variations in wind speed and statistics**

There are not any discernible characteristics of u' in the time domain. The inter-timestep PDFs are all over

the maps. The first one (Case # 1) looks unimodal, the second (Case # 9) is bimodal, the third is negatively skewed, and the fourth (Case # 15) has a relatively flat peak. These PDFs have kurtosis values of various degrees. It may not be possible to guess the distribution solely based on 4S and reconstruct the time series to simulate the inter-timestep variability of wind speed.

Previous studies have established some characteristic features of u' in the frequency domain. Kolmogorov's

hypothesis led to a universal form for the energy spectrum, known sometimes also as the 5/3 rule. There



are power spectrum models listed in IEC 61400-1 (IEC, 2019) including Karman and Kaimal models. The power output of a wind turbine or wind farm seems to possess some unique characteristic signature in terms of the power spectral density (PSD) (Apt, What has been learned from frequency-domain analysis of wind and solar power, 2019). We utilize this characteristic and reverse engineer the time series using the inverse Fourier transformation. The input to the data synthesis application is a 10 minutes time series along with 4S and the output of the data synthesis method is a 1-second resolution time series of wind speed or wind power dataset.


# 3   Literature Review

Recreating actual time series from statistics is an indeterminate problem. However, we can synthesize time series for practical engineering applications. The method of synthesis could be bounded in the time domain or frequency domain or a combination of artifacts from both domains. Some tools can synthesize wind speed in space and time. TurbSim (Jonkman, 2009) utilizes stochastic methods based on the spectrum to generate a wind velocity field across the rotor plane for aeroelastic study. Many time-domain methods rely on the central moments. There is the moment-based Monte-Carlo (MC) (Zhao et al., 2002) and Markov-Chain Monte-Carlo (MCMC) time series synthesis. However, wind industry-standard data only reports the first two moments – mean and standard deviations. For intra-timestep, the statistic of interest is $3^{rd}$ order (skewness) and $4^{th}$ order (kurtosis) central moments. There are only a few data loggers (e.g. NOMAD by SecondWind) that also report skewness and kurtosis.



There are two classes of time series synthesis: interpolation (estimating within values) vs extrapolation (forecast). Such synthesis could be informed either by physical process or by some temporal statistics. The physical approach uses atmospheric/meteorological data to capture non-stationarity whereas statistical approaches are based on the statistic of time series such as moments, PDF, CDF, etc. The statistical approach can generate an instance of time series as well as scenarios representative of the statistics.


HOMER (HOMER, n.d.) uses an academic exercise to generate synthetic wind speed data without many site-specific considerations. It does not resolve the timescale relevant to the power system study. Hybrid2 (E.Baring-Gould, 1996) uses the MC method based on the transition probability matrix (Kaminsky et al., 1990).


Time series models based on Box- Jenkins methods (Browna et al., 1984) are being used for the simulation of wind speed/power for a while but with partial success. Hybrid methods combine the time series technique (ARIMA) with the frequency-domain-based technique. A time series model may furnish parameters for the spectrum (Broersen & Waele, 2013). Once the spectrum is known, we can generate an ARMA process, but the turbulence model requires an order of the model quite high (~1000). Data-driven technique (Stengel et al., 2020) have been recently popular, but these methods can be as good as training datasets. The scope of these methods is limited to a coarse temporal (Daily to Hourly) scale and such datasets are suitable mainly for regional grid integration study and planning purposes.



Frequency-domain-based methods use PSD and wavelets. These methods can help identify some useful information about seasonal (Chen & Rabiti, 20217), and cyclic components of the time series. The wind industry has used PSD to get a better sense of the variation of wind resources and power. Studies (Apt, The spectrum of power from wind turbines, 2007) have revealed some unique characteristics of wind power aggregation. The variable renewable resources seem to have some unique characteristic signatures visible only in the frequency domain. Turbulence characteristics in wind speed approximated by spectrum proportional to $f^{-5/3}$ (Reiteration of turbulence spectrum of wind). The PSDs for three renewables (Solar PV, Wind, Solar Thermal) are similar at very long frequencies (periods greater than about 6 hours, $4 \times 10^{-5}$ Hz),




but there are significant differences at shorter timescales (Apt, What has been learned from frequency-domain analysis of wind and solar power, 2019). The PSD signature is supported by several studies and utilized to understand variability (and uncertainty). Lee (Lee & Baldick, 2014) used piecewise Affine Function Approximation of the PSD (Lee & Baldick, 2014) to synthesize future wind power. There is a hybrid method (Rose & Apt, 2012) to synthesize long-term data at 1 Hz that utilized both the measured and simulated data. This method, however, proposes resampling of the 10 minutes data at 1 Hz using linear interpolation which does not work for wind speed time series as shown in Figure 1. We need a better method to capture intra-timestep variability. We are interested in a practical method simple enough but still can capture variability associated with the aggregated wind and solar power (Piwko et al., 2012) for integration studies, battery sizing, and likewise. We utilize the unique frequency-domain signature, PSD specific to the time step, and synthesize high-resolution (600x) time series by inverse Fourier Transformation. The method we propose automatically encapsulates some physics of turbulence (spectrum, TKE, similarities of scales) by design.

## 4   Methodology

Wind resource measurement typically reports four statistics, 4S = [ $\underline{u}$, σu, u(max), u(min)] of time series u(k) over some aggregation period T. Wind speed is normally sampled at 1 Hz and aggregated at T = 10 minutes or an hour, following IEC 61400-1.

We convert the time domain wind speed signal u(k) to frequency domain u(n) using the FFT. The spectrum in the frequency domain is

$$F_u(n) = \sum_{k=0}^{N-1} \left[\frac{u(k)}{N}\right] e^{-i2\pi nk/N} \qquad (1)$$

Assuming u(t) = x(t) is ergodic, the power spectral density can be defined as:

$S_{xx}(f) = \lim_{T\to\infty} \frac{1}{T}|\hat{x}_T(f)|^2 = \hat{R}_{xx}(f)$ ; where $\hat{x}_T = x(t)\delta(t)$, $\delta(t)$ is the Kronecker delta function, and $\hat{R}_{xx}$ is the autocorrelation function. Using Parseval's theorem, the variance (average power) can be calculated by integrating the power spectrum over all frequencies of interest. The power spectrum $S_{xx}(f)$ is converted to dB/Hz by changing the scale of PSD to 10 log$_{10}$ $S_{xx}(f)$. We use the one-sided frequency spectrum PSD(f) because the wind speed is a real-valued function and the dataset used was sampled at equal intervals.

The simulation of the random processes such as horizontal component u(t, x) of the wind velocity can be described as (Shinozuka, 1971; Veers, 1983):

$$u(t, x) = \underline{u}(t) + u'(t, x) = \underline{u}(t) + \sum_{j=1}^{n} A_j \sin 2\pi f_j t + B_j \cos 2\pi f_j t \qquad (2)$$

Here $A_j$ and $B_j$ are Fourier coefficients corresponding to frequency $f_j$. The wind energy industry uses sampling at a regular interval (1Hz), hence the second term of Equation (2) can be synthesized using the Fast Fourier Transformation ($\mathcal{F}$) as

$$u(t, x) = \underline{u}(t) + \mathcal{F}_{u'}^{-1} \sum_{j=1}^{n} (A_j + i\, B_j). \qquad (3)$$

Our contribution to this literature is that we present an efficient engineering method to recreate the PSD, a function of the Fourier coefficients, practically from the four statistics 4S the wind energy industry reports of the wind resource or wind turbine performance measurement. We recreate PSD by decomposition as follows.



## 4.1 PSD decomposition

We decompose the PSD of u(k) into trend and random components.

$$PSD = PSD\ (T) + PSD\ (R) \tag{4}$$

Analyzing the measured data, we establish that PSD (R) can be estimated using stochastic techniques with an appropriate choice of PSD (T). We present two cases of PSD (T) that use the regular and constrained optimization passing through a fixed point (PSD(Max), fi). The trend PSD (T) and the random component PSD (R) are estimated as follows.


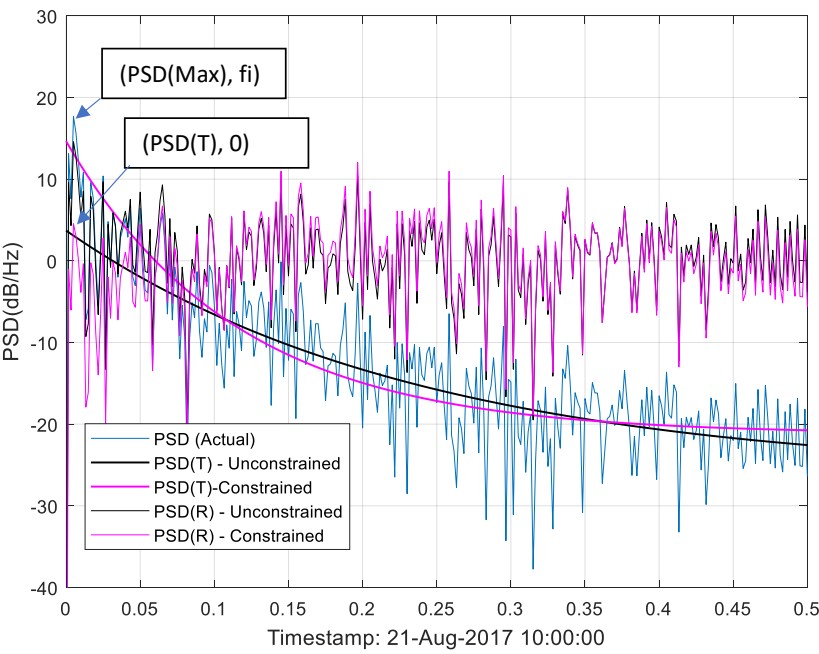

**Figure 2: Power spectral density decomposition**





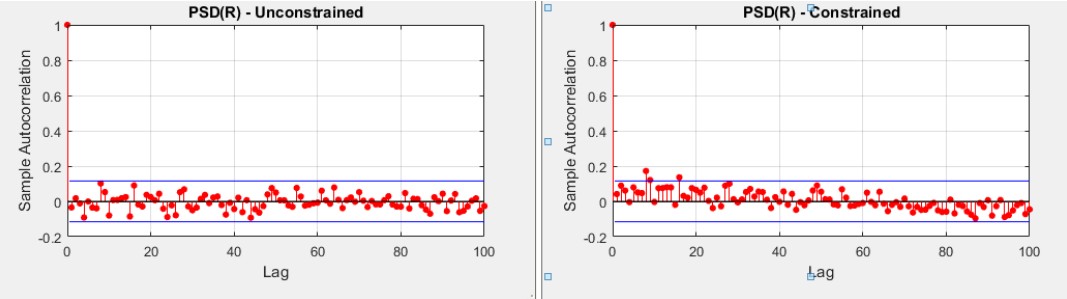

**Figure 3: Sample autocorrelation of PSD(R)**

Looking at the autocorrelation function in Figure 3, the regular (unconstrained) method of estimating the trend PSD(T) leads to a better estimate of PSD(R) as a stochastic process. It is however difficult to estimate the corresponding PSD(at f $\cong$ 0) of the best-fit trend line from the four statistics 4S. A few larger order moments, beyond those included in 4S, might be necessary
to estimate PSD(at f $\cong$ 0).

## 4.2 Trend of PSD Estimate

The power spectrum is distributed exponentially for a time series that is formed from a sequence of measurements of a normally distributed random variable with mean zero and variance unity, N(0,1), [20]. Accordingly, we estimate the trend component of PSD by an exponential function:

$$\text{PSD (T)} = a_1 \, exp^{a2*n} + a_3;  \tag{5}$$

where ai (i= 1, 2 3) are parameters, and n is the frequency in Hz.

We use both the regular (unconstrained) and constrained least square best-fit methods to estimate ais. In the case of the latter method, PSD(T) must meet additional constraints. The PSD(T) has to pass closely through a fixed point (PSD(Max), fi) which is an additional constraint of the non-linear optimization to
come up with the parameters (ai). We choose the constrained method option for PSD(T) because the one of inputs, the standard deviation, is the direct predictor of the PSD(Max) as presented in Figure 4, which may be following Parseval's theorem.



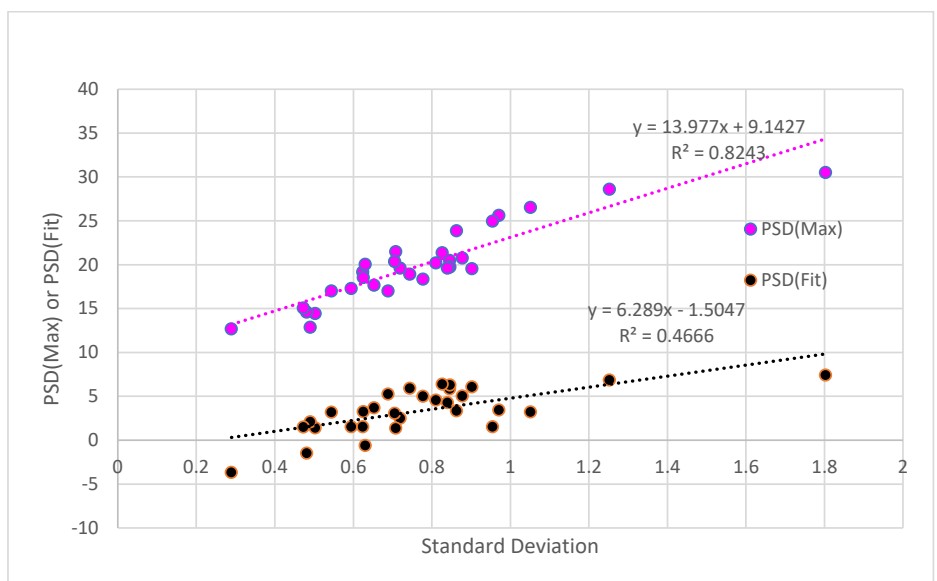

**Figure 4: Relationship between PSD(T) and Standard Deviation**

## 4.3 Random Component of PSD Estimate

The random components PSD(R) are estimated based on the lower and upper envelope of the PSD(Trend). Based on the analysis of measurement data, we used an envelope of the PSD as [PSD (T) + ue, PSD (T) - 2*ue]. We are arbitrarily using ue = 10 by the hit and trial methods.

We model PSD(R) as a stochastic variable. A random number r was generated as follows:

$$r = lb + (ub-lb).* randn(N,1); \text{ with } lb = -ue, ub = ue;$$

We then rescaled r with constants [ks, ke] = [0.3, 0.4] as follows.

$$
\begin{aligned}
r \quad &= (r - ue)*ks &&\text{if } |r| > ue \text{ and the sign of r is positive,} &&(6)\\
&= (r + ue)*ke &&\text{if } |r| > ue \text{ and the sign of r is negative}\\
&= r &&\text{otherwise}
\end{aligned}
$$

Figure 5 presents PSD(R) derived from the uniformly distributed random numbers RAND(). For comparison, we have overlaid the synthesized PSD(R) with the derived PSD(R) from the CART wind turbine.

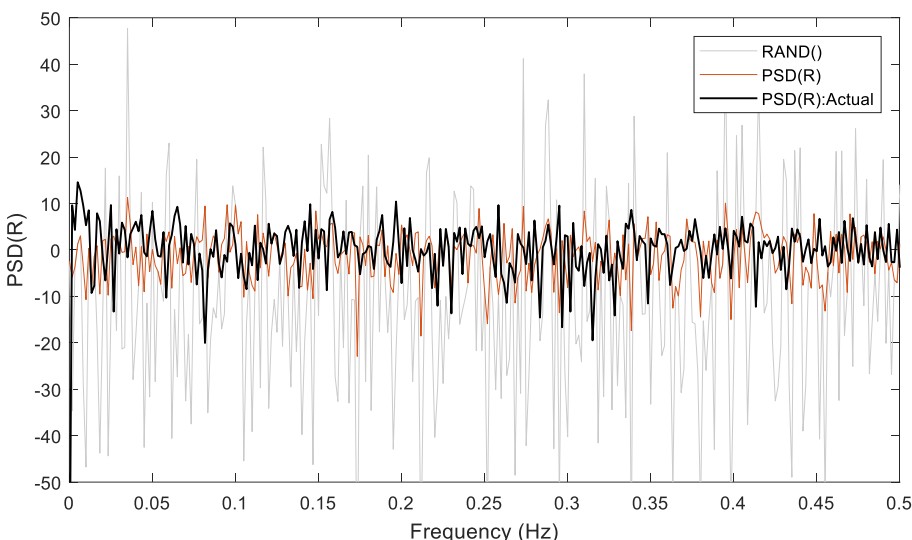

**Figure 5: Actual and Modeled PSD(R) derived from the random numbers**

The range (or envelop) of PSD(R) seems to have comparable magnitude and randomness with the actual data.

## 4.4   Data synthesis

Here we present data synthesis for the t = 10 minutes, and t > 10 minutes for which our proposed method requires addressing discontinuity at the border of the two timestamps.

**A)  10-minute time stamp at f = 1 Hz**

The PSD is now constructed following Equation 4. The PSD is used to synthesize the time series using the inverse FFT.  The transformed output signal (from ifft) was rescaled to the standard deviation and extremums of u' based on 4S.

**B)  Connecting two timestamps**

The method described above helps synthesize 1-second resolution data for 10 minutes from one timestamp say at t. To create a time series spanning the multiple timestamps, we need to repeat the process for t-1, t timestamps and juxtapose them in the order of the timestamps.  These timestamps, however, are synthesized without considering previous (t-1) or succeeding (t+1). Hence it is natural to anticipate the discontinuity at the border. To synthesize data for a duration T > 600 sec (i.e. 10 minutes) we must address the discontinuity

at the border where two 10-minute timestamps join. We connect these two consecutive timestamps using the simple algorithm presented in Section 8.2.

## 4.5   Metrics for time series comparison

The method we propose here does not resolve the phase angle among various frequencies comprising the time domain signal u(t, x). Instead, we use a randomized phase angle sequence while creating time series

from the PSD.  Accordingly, this PSD decomposition technique is to capture the variability of wind speed



not the temporal accuracy of the variations within the time T = 10 of synthesis. We rescale the synthesized u'(t, x) to match the required standard deviation σu of 4S input. Hence, the coefficient of variation which is the ratio of σu/ u matches well between synthetic and actual data by design. We can also use the root mean squared (RMS) value as a metric to compare $\hat{y}$ with y. The RMS value, however, does not add much information a time series of u' scaled to the targeted σu.

An alternative metric to compare time series is the goodness of fit (GFIT). Here GFIT is computed of the actual time series (y) and synthesized time series ($\hat{y}$) using Equation 7, which is based on the L2 -norm:

$$\text{GFIT} = \left[1 - \frac{NORM(y-\hat{y})}{NORM(y-\bar{y})}\right] \times 100\ \% \qquad (7)$$

For time-series comparison it is ideal to compare the higher-order central moments: $E[(u - E(u))^n]$. We know the synthetic time series has lost its temporal accuracy as a result of the randomized phase introduced during the FFT. Accordingly, the metrics for the test of reverse-engineered time series can't be the higher-order central moments. Hence, we use the two-sample Kolmogorov-Smirnov test (KS2) that infer if time series data are from the same continuous distribution. The test statistic result h is 1 if the test rejects the null hypothesis (Ho: the two time series data are from the same continuous distribution) at the 5% confidence level, and 0 otherwise. p is the probability the test statistic h is as extreme as the observed value under the null hypothesis.

# 5 Measurement and Validation Dataset

We use an archived dataset of CART3 at the NREL's Flatirons Campus. The original data for CART3 was collected using LabView DAQ at 400 Hz and involves over 50+ channels capturing the state of the wind turbine and corresponding wind resource measurement synchronized over the GPS clock. We aggregate the measured 400 Hz field data at T = 1 sec, 1 minute, and 10 minutes and illustrate the concept underlying the intra-timestep wind speed synthesis.

The second dataset used in this study is from NPS 100-21C (Broyhill Wind Turbine) at Appalachian State. This 100 kW wind turbine manufactured by the Northern Power System (NPS) is in operation since 2009. We use one day of rolling operational data sampled approximately at 1 Hz.

Table 1: Metadata of the two datasets used for validation.

| Dataset | Site Name | Location | Sampling Frequency | Measurement Hub Height | Data Span |
|---------|-----------|----------|--------------------|-----------------------|-----------|
| CART3 | NWTC, NREL | Aurora, Co | 400 Hz | 36.6 m | 1 Day ( August 21, 2017) |
| NPS100 | Broyhill, Appstate | Boone, NC | ~ 1 Hz | 37.4 m | 1 Day (March 30, 2022) |

# 6 Parameters estimation for PSD

We present some analytical equations to estimate parameters ai (i= 1, 2 3) from 4S. These equations are based on the correlation observed in the field data (400 Hz). We aggregated the CART3 field data at 1 second and computed 10 minutes statistics following the industry standard practice stipulated in IEC 61400-1. Figure 7 presents PSD(Max) plotted against σu, and the slope $a_2$ of PSD(T) with $a_1$. At frequency n =0, from Equation (4), PSD(T) = $a_1$ + $a_3$. This is the maximum value of the best fit exponential function PSD(T) which is different from PSD(max).




Based on empirical evidence present in measured data, there is a linear relationship between $\sigma_u$ and PSD(max) as illustrated in Figure 4:

$$\text{PSD(Max): } y = 13.977\,\sigma_u + 9.1427 \quad [\text{with } R^2 = 0.8243] \tag{8}$$

$$\text{PSD(Fit): } y = 6.289\,\sigma_u - 1.5047 \quad [\text{with } R^2 = 0.4666] \tag{9}$$

We found a better correlation of PSD(Fit) with the extremums. Figure 6 presents a chart of
PSD(Fit) as a function of $[(u(max) - u(min))]^n$, where n = 1, ½ and ¼ .

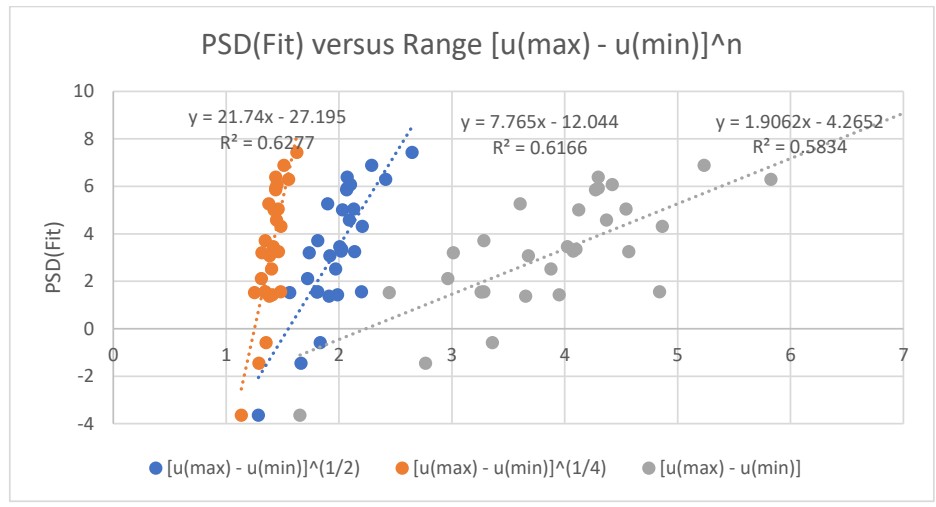

**Figure 6: PSD(Fit) as function of range of u(max) and u(min)**

Accordingly, we used the following relationship to estimate PSD(Fit) in this paper,

$$\text{PSD(Fit): } y = 21.74[\Delta(u(max) - u(min))]^{1/4} - 27.195 \tag{10}$$

The slope ($a_2$) of the PSD (Trend) is found to be related to $a_1$ by a power relationship as:

$$a_2 = 1993.9\,a_1^{-1.793} \quad [\text{with } R^2 = 0.7713] \tag{11}$$

Also, $a_2$ relates to $a_1 * a_3$ as following an exponential relationship as below:

$$a_2 = 7.1356e^{0.0006(a_1 * a_3)} \quad [\text{with } R^2 = 0.7439] \tag{12}$$

and, $a_1$ and $a_3$ are related to each other by a linear relationship.

$$a_3 = -1.0752\,a_1 + 6.0873 \quad [\text{with } R^2 = 0.8716] \tag{13}$$

These empirical relationships among the parameters are depicted in Figure 7.





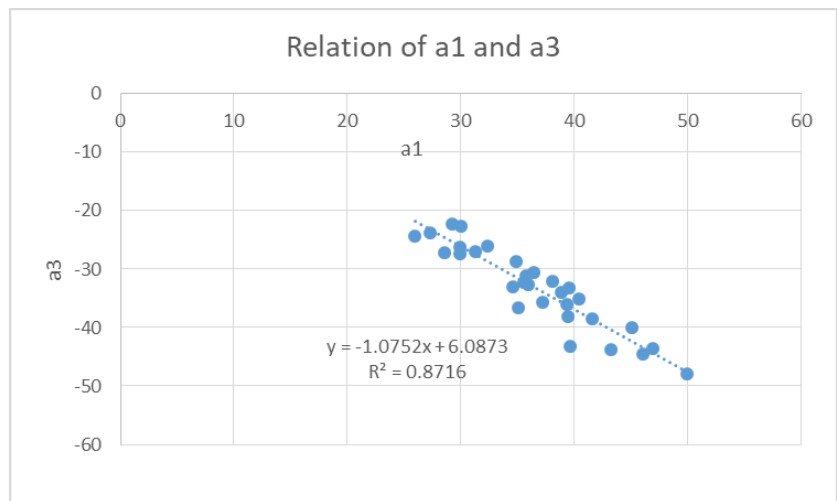

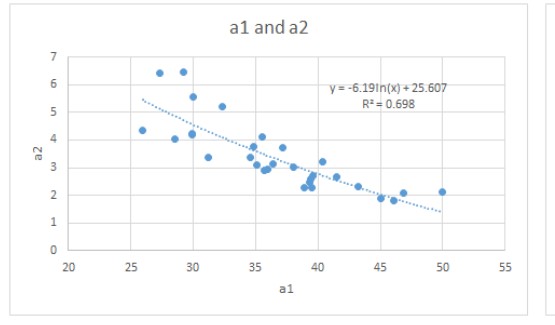
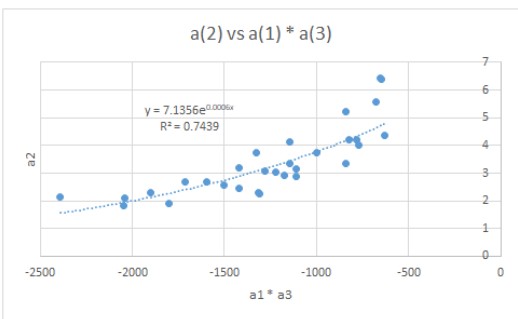

**Figure 7: Parameter estimation and mapping**

# 7 Steps to synthesize a time series

In this section, we present steps to synthesize a 10-minute time series using the empirical relation we propose among various parameters comprising the PSD function. It is assumed that we have 4S (at 10 minutes) for the timestamp for which we want to synthesize 1 Hz wind speed or wind power data. Here are the sequential steps we propose to reverse engineer the time series.

1) Estimate PSD(Fit) following Equation 10. PSD(Fit) is a measure of $a_1 + a_3$. You can also use the standard deviation to estimate PSD(Fit) as presented in Equation 9.

2) Determine the exponent $a_2$ using Equation 11.

3) Estimate the trend: $PSD(T) = a_1\, exp^{a_2*n} + a_3$



4) Model the PSD(R) as described in Section 4.3.

5) Sum these two functions; PSD = PSD(T) + PSD(R)

6) Convert the one-sided PSD in (5) to two-sided PSD, say PSD2.

7) Generate a random number between [0 1] from the uniform distribution.

8) Use iFFT to generate the time series of u′, u(k) = ifft[fft(rn).*sqrt(PSD2)];

9) Scale the u(k) to zero mean and extremums first and then to the targeted standard deviation.

10) Make sure the range of u(k) confirms the extremum values of u′. Replace any outliers of u(k) by u′ (max) or u′ (min) as necessary and repeat step 9 to confirm the targeted standard deviation. You should now have u'(t, x) of Equation (2).

# 8   Results and Discussion

## 8.1   Time series synthesis

Figure 8 compares synthesized time series for one of four cases illustrated in Figure 1 of Section 2. This is case # 15. By observation, we may tell that the variability of the wind speed looks comparable. A pitfall of the method is that the two dataset does not compare well in the temporal dimension. This deviation may be attributed to the lost phase information of frequency components comprising the signal. We don't know how those frequencies are spaced apart relative to each other.

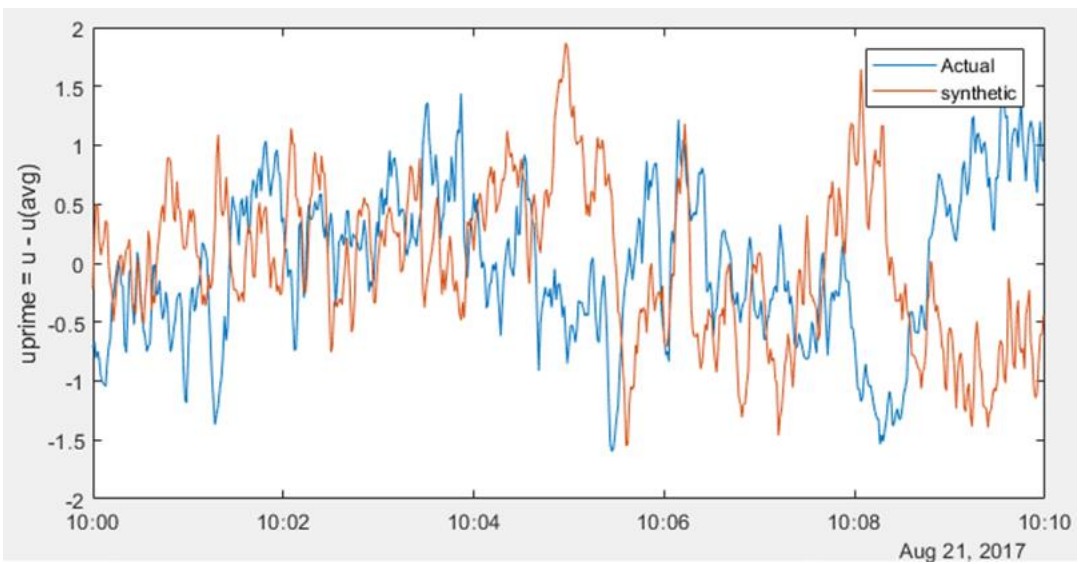

**Figure 8: Actual and synthesized wind speed time series**



## 8.2 Connecting two timestamps

We assume continuity at the border to come up with a simple logic to connect two 10-minute timestamps. Assuming $\underline{u}(t)$ is the 10-minute average wind speed at t and $\hat{u}'(t, i)$ is the synthesized fluctuating component of 10-minute where index i of $\hat{u}'(t, i)$ goes from 1 to 600 for 1 Hz data. Here, $u'_f$ is the first value corresponding to i = 1 and $u'_l$ is the last value i = 600 of the $\hat{u}'(t)$.

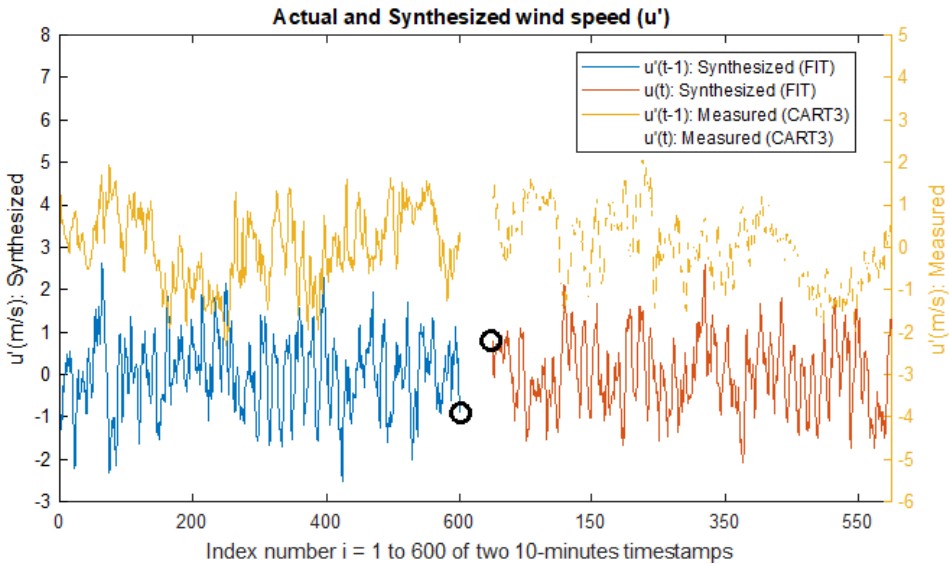

**Figure 9: Discontinuity at the border of two synthesized timestamps**

By its design, this method embodies a discontinuity at the border, see the ends marked by two black circles in Figure 9. To avoid sharp discontinuity at the border, it demands

$$\underline{u}(t-1) + \hat{u}' (t-1, i = 600) \cong \underline{u}(t) + \hat{u}' (t, i = 1)$$

or $\quad \underline{u}(t-1) + \hat{u}'_l(t-1) \cong \underline{u}(t) + \hat{u}'_f(t)$

or $\quad \hat{u}'_f(t) - \hat{u}'_l(t-1) \cong \underline{u}(t-1) - \underline{u}(t)$ $\hfill (14)$

We looked at the distribution of $\Delta u' = u' (t+1) - u'(t)$ to estimate the parameter to connect two timestamps. We present the data from CART3 in Figure 10 and Table 2 presents two distributions, normal and logistic, fitted to the $\Delta u'$ and corresponding distribution parameters.

**Table 2: Distribution parameters of $\Delta u'$**

| Distribution | Normal | | Logistic | |
|---|---|---|---|---|
| parameter | value | standard error | value | standard error |
| mu ($\mu$) | -0.0008 | 0.0015 | -0.0062 | 0.0014 |
| Sigma ($\sigma$) | 0.2061 | 0.0011 | 0.1095 | 0.0007 |

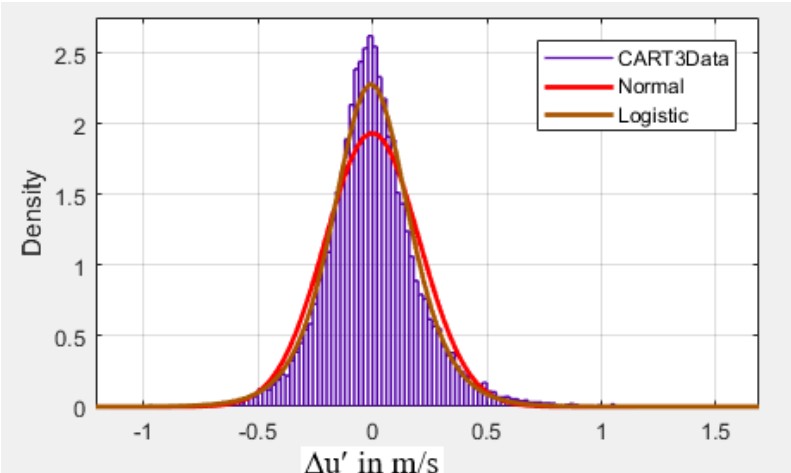

**Figure 10: Distribution of Wind speed difference in 1 Hz data**

Following Equation 14, one can evaluate the difference of 10-minute average wind speed between two timestamps. If the absolute value of the difference between LHS and RHS of Equation 10 is greater than a parameter ε (a function of sigma Table 2), we may use a moving average filter for u′ at the border. The window of the moving average filter is a parameter that can be set based on the magnitude of discrepancy at the border of the two timestamps.

For the result presented in the following section, we checked if $|\Delta u| = |u'_f(t) - u'_l(t-1)| \geq 0.5$, which corresponds to about $\pm 2.5\sigma$ and represents about 80.26% of all measured variations u′ at 1 Hz). If yes, we averaged the two values at the border. For all other cases, the time series are juxtaposed as it is they come out of the inverse FFT and are scaled to the 4S.

## 8.3   Comparison with a similar method

We compare the time series generated by the PSD decomposition method with a method that takes similar inputs. Here, we compare results with the Hybrid method proposed by Rose and Apt (Rose & Apt, 2012) using the Kaimal spectrum where σu is explicit. The industry standard 10-minute averaged statistics 4S should be known apriori for both methods.

We evaluated different filters including the Butterworth method they propose for the slowly varying measured wind speed. The Piecewise Cubic Hermite Interpolating Polynomial (pchip) and the cubic spline
interpolation method seem to match the result they presented (Rose & Apt, 2012) in Figure 1 (c). Accordingly, a comparison presented here uses the "u(T=10-min): Trend(pchip)" value as shown in Figure 11. Please note that the PSD decomposition method uses the measured aggregated value u(T = 10-min) to calculate u(t) whereas the Hybrid method uses the Trend(pchip) which varies within the 10-minute timestamp.



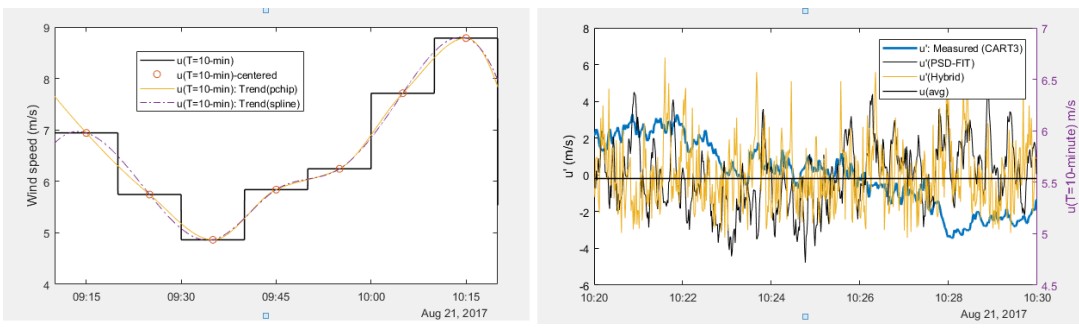

a)  Average value and trend value for comparison        b) Measured and simulated u'

**Figure 11: Comparison of u̲ and u′ for the Hybrid and PSD Decomposition methods**

It was observed that both methods have a possibility of resulting in a negative wind speed, especially for the time stamp where the standard deviation is larger and the mean value is low. The case in point here is presented in Figure 12 (b), the time stamp starting at 10:20 am with 4S = [ u̲ = 5.536; σu = 1.803, u(max) = 9.036 , u(min) = 2.024 ] m/s. Hence, we had to use an additional constraint (i.e. u(t) ≥ 0) while scaling the standard deviation to the measured value σu = 1.803.

Figure 12 compares the synthesized time series for an hour with the measured CART3 data. For a better visual comparison, we present a zoomed-in version of the initial 10-minute timestamp in Figure 12 (b).

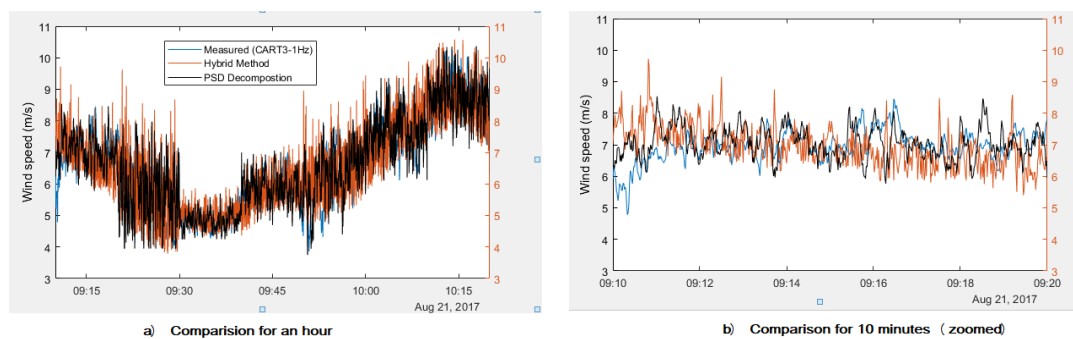

a) Comparision for an hour                                    b)  Comparison for 10 minutes (zoomed)

**Figure 12: Comparison of the PSD Decomposition technique with the Hybrid Method (Rose & Apt, 2012)**

The PSD decomposition method seems to follow the actual data better than the Hybrid method. Figure 13 compares the PDF and CDF of synthetic data by these two methods with that of the measured CART3 dataset. We also compare the PSDs to check how closely this method is reversible. A data synthesis may be called reversible it has both forward and backward compatibility. In our case, the synthetic data should produce the PSDs and vice versa without significant loss of information. This is not the case as presented in Figure 15. There is a significant loss of information while trying to recreate the measured PSD from the method discussed in this paper.

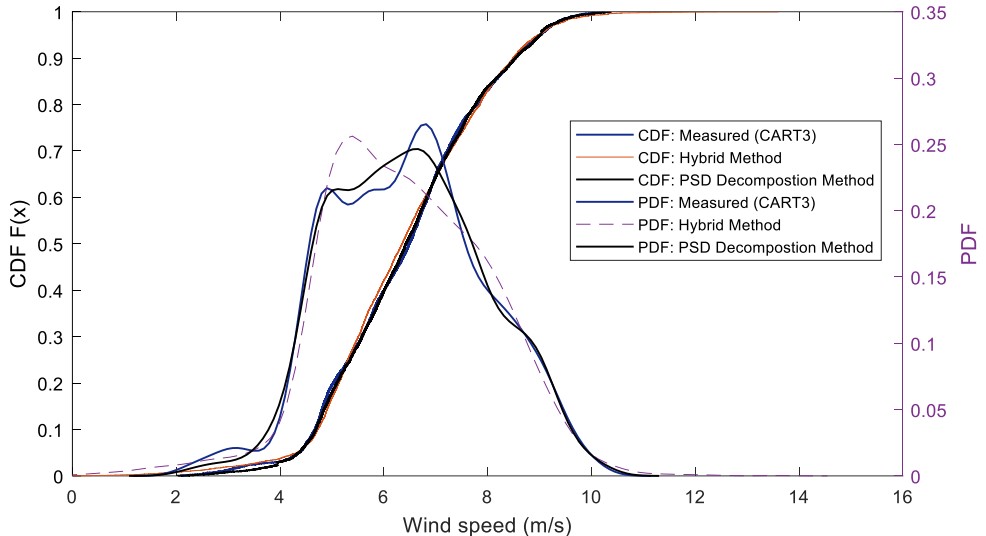

**Figure 13: Distributions of Actual and Synthetic Data.**

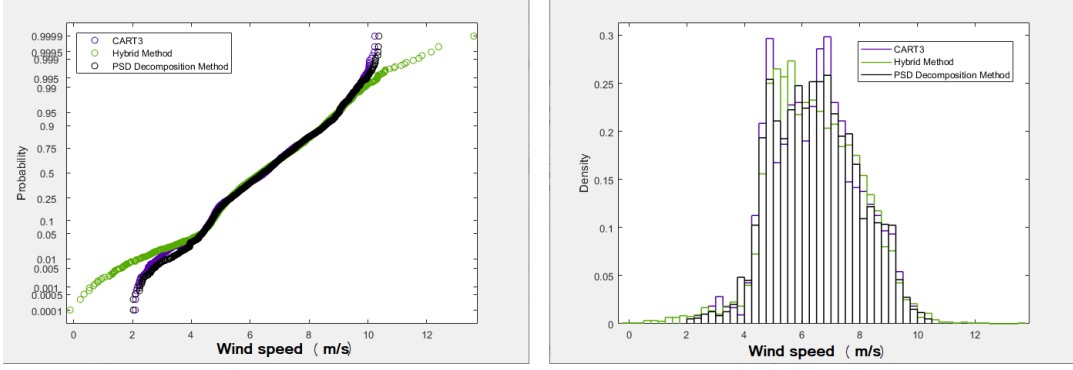

**Figure 14: Probability Plot and Bin-wise Density Function.**


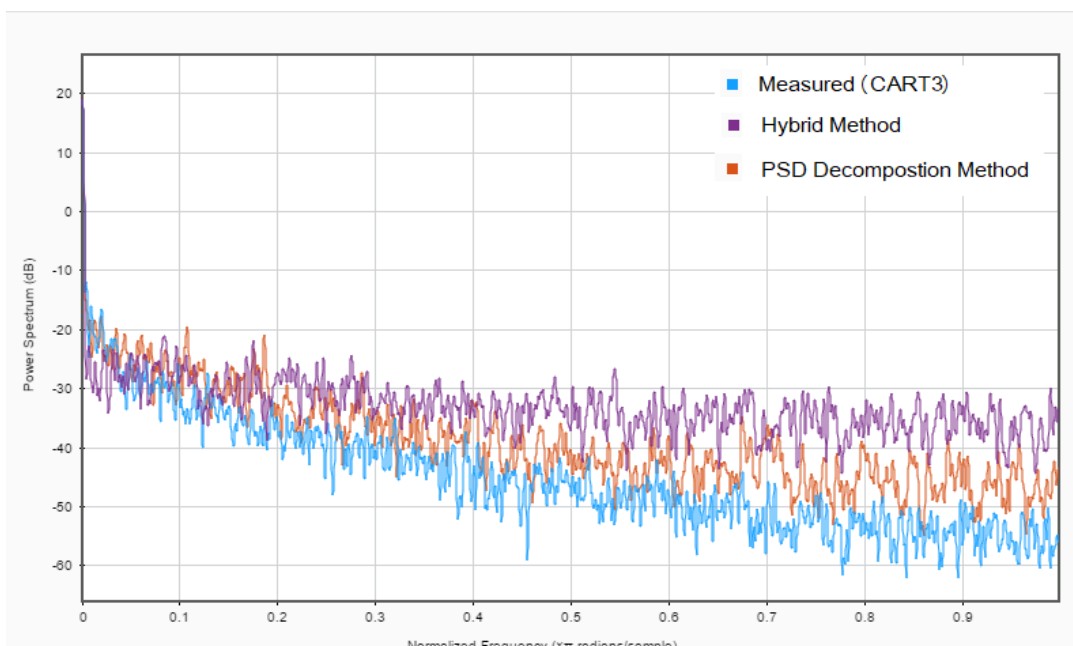

**Figure 15: Power spectrum comparison with the measured CART3 data.**

The comparisons of PDFs presented in Figure 13, the probability plot in Figure 14, and the power
spectrum in Figure 15 may imply that the PSD decomposition method outperforms the Hybrid
method. It is however premature to draw any final conclusion based on the limited dataset used in
this study.

## 8.4   Discussion

In Section 2 we identified a few study cases and assigned them a case number. Table 3 presents these cases
with some statistics to compare the synthesized time series with the measured data. For the two-sample
Kolmogorov-Smirnov test, we present the test statistics h and p-value. The p-value is a scalar in the range
(0,1) which indicates whether the test results are significant. The test statistic result h is 1 if the test rejects
the null hypothesis (Ho: data in the two time series are from the same continuous distribution) at the 5%
significance level, and 0 otherwise.

Table 3: Metrics for the time series comparison

| | | | Measured Data | Synthesized Data | | | | Two Sample Kolmogorov-Smirnov | | | |
| | | | (CART3) | PSD(Fit) | | PSD(Max) | | PSD(Fit) | | PSD(Max) | |
| Case # | DateTime | Timespan | RMS | RMS | GFIT (%) | RMS | GFIT(%) | h | p | h | p |
|---|---|---|---|---|---|---|---|---|---|---|---|
| 1 | 8/21/2017 1:10 | 10 min | 0.869 | 0.878 | -52.572 | 0.880 | -49.695 | 0 | 0.666 | 0 | 0.715 |
| 9 | 8/21/2017 2:40 | 10 min | 1.251 | 1.280 | -32.005 | 1.917 | -91.137 | 1 | 0.000 | 1 | 0.000 |
| 13 | 8/21/2017 9:40 | 10 min | 0.479 | 0.500 | -47.971 | 0.531 | -38.253 | 1 | 0.000 | 1 | 0.000 |
| 15 | 8/21/2017 10:00 | 10 min | 0.648 | 0.666 | -41.457 | 0.653 | -37.910 | 1 | 0.002 | 0 | 0.251 |
| 17 | 8/21/2017 10:20 | 10 min | 1.802 | 1.803 | -55.498 | 1.802 | -21.039 | 0 | 0.076 | 1 | 0.001 |



The RMS values match closely with the measured data because of the rescaling of the synthesized data by the target standard deviation. When μ = 0, which is always true for u' by the virtue of Reynold's decomposition, the RMS value and standard deviation differ slightly because of a difference in the degree of freedom. The negative value of GFIT implies mathematically that $NORM(y - \hat{y}) > NORM(y - \bar{y})$, which physically means that the directions of y and $\hat{y}$ are not well correlated because we used the randomized phase during the inverse FFT. However, the two-sample Kolmogorov-Smirnov test suggests that the method we propose is able to recreate the distribution of the measured data for Cases # 1, 15, and 17. The PSD decomposition method seems not to capture the variation of u' if it is bimodal (Cases # 9) or heavily skewed positively or negatively. This result suggests that we need to measure some additional parameters, higher order moment functions such as skewness and kurtosis or phase sequence of the underlying FFT, to completely inform this 1 Hz data synthesis process.

Given the technological advancement in data acquisition and data storage, does it worth exploring the next optimal timestep to archive/aggregate wind data? Is the current practice of 10-minute aggregation good enough or should we aspire for data aggregation at a higher resolution (say for example, 1-minute or 5-minute)? These questions may demand a larger discussion in order to attempt consensus among diverse stakeholders of the wind energy industry. As part of the broader research questions, we proposed a temporary fix for data synthesis, from 10 minutes to 1 second, to capture inter-timestep variability of wind speed or wind power. The PSD decomposition method we detailed in this article seems to be a practical solution but not general enough to capture a wider range of distribution u' may follow. We aspire to come up with additional parameters based on data sampling of wind resources and wind power that might complement the wind energy industry-standard 10-minute aggregation practice.

# 9 Conclusion

We present an engineering method to synthesize a high resolution ( f = 1 Hz) wind speed time series based on the industry standard four standard statistics (i.e. average, standard deviation, maximum, and minimum) recorded at 10 minutes. We utilize the unique signature embedded in power spectral density (PSD) to synthesize a high-resolution (600x) dataset. This data synthesis method by PSD decomposition encompasses some physics of horizontal wind speed inherent in the power spectral density signature by its design but this method is not general enough to approximate all possible variations of u' within the 10-minute aggregation time.

We have not explored the forward and backward compatibility of the PSD decomposition technique for data synthesis. Future works may pinpoint what additional parameters we may need to collect during the wind resource assessment phase for an accurate representation of the spectrum and facilitate interoperability of data synthesis such that industry-standard 10 minutes data can still be utilized to inform applications requiring high-resolution data such as grid integration and power system studies.

## Competing interests

The contact author has declared that none of the authors has any competing interests.

## Acknowledgment

Andy Scholbrock (NREL) and Chris Conner (NPS) for their support with the validation dataset. We benefited from comments we received at the 2022 NAWEA/WindTech Conference at the University of Delaware.





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
