# Peer review of "Wind speed time series synthesis using a parametrized power spectral density function"

_Wind Energy Science, 2023_

## Author Comment (AC1)

**Response to Reviewers Comments: wes-2023-45**

Dear Editor:

Thank you for all these valuable comments. Obviously, these comments help improve the clarity of idea/concept, organization of write-up, and eventually the quality of the manuscript. Our responses to specific comments are appended below (see attached PDF file).

Please let us know if you have any other comments and suggestions to improve the revised manuscript. Thank you very much!

*Ram Poudel*

======================Reviewer Comments=============================

Overview

This manuscript proposes a statistical method for reconstructing 1 Hz wind speed and power time series using the 10-minute wind statistics. The method is based on a representing power spectral density function via the so-called trend and random components. However, this concept is not clear to the reviewer and is not properly described in this manuscript. Furthermore, the manuscript is not properly formatted (there are too many sections) and the quality of the figures is not satisfactory.

Specific comments

1. A number of references are in capital letters. Is there a reason for that?

**Response:** This is a random error in reference style, we have corrected in this revision.

2. Mathematical variables should be italic and written using an equation editor (e.g., Lines 31, 50, 51, 53, 54, 55, etc.).

   **Response:** I did not know these rules of mathematical variables earlier. We have updated this formatting error in this revision, including Eq 2 and Eq 3 redone with equation editor.

3. Figure 1. This figure is too small and it is very difficult (border impossible) to read labels, properly see time series, etc.

**Response:** Thank you, we have redrawn this figure and tried to make labels better legible.

[Figure]

4. Figure 1. What instrument was used to collect these 400 Hz data? At such a high acquisition frequency, I would expect more fluctuations in the wind speed. What is written on the *y*-axis? What is the orange (or red) line in the top row?

**Response:**

==== =====A reply from Andy Scholbrock (NREL) regarding the instruments used==============

The wind speed is either measured by ATI Sonic "K Style" anemometer (https://www.apptech.com/probe-styles/, and manual attached), or by our MetOne WS-201 Cup anemometer (manual attached).

 Regarding the data frequency, although we record everything at 400 Hz, some systems don't provide that high of frequency of data.  For the cups, I wouldn't expect to get any useful information beyond 2 to 3 Hz.  The sonics I would think are closer to 20 to 30 Hz.  Reading through the attached manual for the sonic, looks like the ATI is closer to 10 Hz.  So, our DAS system oversamples these sensors and hence you don't see higher frequency content in a time-series plot.

========================================================================

A. Scholbrock, P. Fleming, L. Fingersh, A. Wright, D. Schlipf, F. Haizmann, and F. Belen. Field testing lidar-based feed-forward controls on the NREL controls advanced research turbine. In Proc. AIAA Aerospace Sciences Meeting, Grapevine, TX, January 2013.

The data was collected using methods and instruments mentioned above. We have aggregated the 400 Hz data to 1 Hz, and presented in Figure 1, hence you may not see more fluctuations – we did aggregate to make it comparable to DAQ resolution wind energy industry uses.

5. The –5/3 law by Kolmogorov (not 5/3) only holds for the inertial subrange where the average properties of the turbulence are governed by the rate of dissipation. So, his model does not cover the entire energy spectrum, as your L70 might indicate.

**Response:** True. We have updated the sentence to include inertial subrange and corrected it to -5/3. This is just a mention of related literature.

6. Broken reference.

**Response:** I realized there are issues with reference style when I changed from original IEEE style to APA. We have tried to establish the broken reference. Hopefully, typesetter will take care of any remaining formatting/reference issues.

7. 10-5?

**Response:** L113: (4x10−5 Hz) – updated to ($4 \times 10^{-5}$ Hz)

8. L114–115. What does this stand for: "Apt, What has been learned from frequency domain analysis of wind and solar power, 2019?" Is that a statement or a very long reference?

**Response:** This is a reference style error; we have corrected it in the revision.

9. Equation 1. Explain all variables in this equation and subsequent in-line equations (e.g., $N$, $n$, $k$, $t$, $i$, etc.). While some of them might be obvious, they should still be formally defined.

**Response:** Here is what we added immediately after Equation 1.

where, $N$ = length of the vector data; $n$ = frequency $\leq$ Nyquist frequency; $k$ = time index; $i$ imaginary number such that $i^2 = -1$; and $e^{-i2\pi/N}$ is $N^{th}$ root of unity.

10. There should be a space between the number and the unit.

   L145: (1Hz), $\rightarrow$ (1 Hz)

11. Figure 2. There are two black lines in the graph. Which one is PSD(T) and which one is PSD(R)?

**Response:** We put the same black color to show they belong to the same PSD decomposition. To avoid confusion, we now have added two boxes and two arrows pointing to PSD(T) and PSD(R) in Figure 2 – see added insets as well.

[Figure]

12. Figure 2. If the authors are not using the evolutionary power spectral density, then how the PSD in Figure 2 (*y*-axis) can be time-dependent (*x*-axis)?

**Response:** We have decomposed PSD = PSD (T) + PSD (R). PSD(T) is parameterized with 4S (average, stDev, maximum, minimum). 4S is a function of time which evolves every 10-minute following the wind energy industry standard practice of data aggregation. Hence, it is equivalent of using piecewise evolutionary PSD – the PSD(T) evolves every 10-minute time window.

PSD(R) is modelled using stochastic processes/techniques.

13. Discussion around Eq. (4) and this whole method are not clear to this reviewer. Perhaps they are sound, but the authors need to do a better job at explaining the concept they are introducing. For example, this discussion does not properly

explain the difference between constrained and unconstrained power spectral densities, and Figure 3 further complicates this whole issue.

**Response:** Thank you for pointing this out. We have updated Section 4.1; changes are marked in a different color in this revision. Hope the method is clear now. Please let us know if you want to include further details.

14. Equation (5). What is the source of this equation? If the authors came up with this expression, what data support it? exp is a function and should not be italic.

**Response:** The authors came with Equation (5). The extent of data support for this equation is presented in Figure 2, Figure 3 and Figure 15.

Yes, exp is a function, we removed Italic format.

15. ais?

**Response:** This confusion may be due to the missing subscript.

$a_i$s --> $a_1$, $a_2$, $a_3$ etc; as defined in previous sentence.

16. What is ue?

**Response:** Consider a buffer around PSD(T) as illustrated by upper and lower buffer lines in figure below. ue is the distance between upper buffer line and PSD(T). It was found the lower buffer is about twice (2 * ue) of the upper buffer.

By observation, we have set ue = 10.

[Figure]

Timestamp: 21-Aug-2017 10:00:00

The envelop/spread of actual PSD is found about 1 unit above and 2 units below PSD(T). ue quantify this spread of PSD. We choose to use ue = 10. 1 unit = 10 dB/Hz.

17. If data are not statistically stationary, the proper spectral analysis is the evolutionary power spectral density and not the regular power spectral density. The authors should comment on how this statement relates to their "trend" power spectral density estimate.

**Response:** This is a great point we missed earlier. We updated the manuscript (see Section 4.1) as follows.

We decompose the PSD of u(k) into trend PSD (T) and random components PSD (R). The additive model of PSD is given by,

$$PSD = PSD\ (T) + PSD\ (R). \tag{4}$$

We approximate PSD(T) by an exponential function. PSD(T) changes every 10 minutes following the industry-standard 10 minutes data aggregation. PSD(T) is estimated using empirical correlations based on 4S. Accordingly, PSD evolves over time following 4S. This method uses evenly spaced piecewise evolutionary power spectral density to synthesize non-stationary data.

The wind energy industry samples data at 1 Hz and aggregates the sampled data every 10 minutes. By the nature of the problem we are trying to solve here, PSD (T) evolves every 10 minutes. The 10-minute data is implicitly assumed to be statistically stationary and represented by 4S. This can be considered as a **piecewise** statistically stationary but evolving PSD every 10 minutes.

18. Section 4.3. This section is not clear to me and how does it contribute to, for example, Shiznozuka (1971) method (e.g. Equation 2 in this manuscript)?

**Response:** To use Shiznozuka (1971) method, we need to know Ai and Bi of Equation 2. Ai and Bi are constituents of PSD or S(f).

We estimate PSD(T) as function of 4S – established using empirical correlation from measured data. We get Ai and Bis by the Inverse Fast Fourier Transformation of the PSD.

Input: 4S $= [u, \sigma_u, u(max), u(min)]$   |   Output: PSD

[Figure]

I hope the figure above clarifies how it links to Shizouka method and eventually to time series synthesis.

19. The quality of the figures in the rest of the manuscript is low. Some figures have grey regions outside of the graphs, which is not the standard publishing practice in journals.

**Response:** I have redrawn Figure 1. We have removed grey regions from Figure 4, Figure 6, and one of Figure 7. The grey regions came as I took screenshots from MATLAB and Excel, I have cropped out some of them where feasible in this revision.

20. Overall, I find this manuscript very difficult to follow. The authors have too many sections—there are 9 sections and most of them have multiple subsections.

**Response:** We hope this revision is better. We have incorporated most of your suggestions and feedback. Your comments helped improve the quality of the paper and its flow/organization.

21. I do not see how this method contributes to the body of literature compared to TurbSim or other methods that are used to reconstruct fluctuating wind components from mean speed and standard deviation.

**Response:** TurbSim offers the Kaimal (IECKAI) and von Karman (IECVKM) spectral models as defined in the IEC standard IEC 61400-1, Ed. 2/Ed.3. The models used in TurbSim and IEC are more general and also capture turbulence across space (x, y, z), and spatial coherence.

For example, the equation below is Kaimal (IECKAI) model for spectrum.

$$S(f) = \frac{4\sigma^2 L/u}{(1 + 6f \cdot L/u)^{5/3}} \tag{2}$$

where $f$ is the cyclic frequency and $L$ is an integral scale parameter dependent on the hub height. $u$ is the mean wind speed at hub height. $\sigma$ can be estimated by the turbulence intensity $TI$ (%)

$$\sigma = \frac{TI}{100} u \tag{3}$$

Kaimal model has an analytical expression for S(f) whereas our model is based on empirical correlations. S(f) [Kaimal] = F(L, u, σ), whereas S(f) in our method is F(4S).

Both methods give spectrum S(f) but input parameters are different. Our method uses 4S which are typical measured inputs/statistics. We have come up with practical engineering method estimate S(f) in order to synthesis time series of horizontal component of wind speed.

We can study how these models compare in a separate subsequent study. We have revised conclusions to incorporate these insightful comments.
* * *

---

## Author Comment (AC2)

**Response to the Reviewer 02 Comments:**

We benefited greatly from two reviewers looking at two different aspects of this manuscript. The first reviewer on technical and the second one on editorial plus technical aspects. We concur with both the reviewers' comments and have revised the manuscript accordingly. I believe these comments helped improve quality of the manuscript which we have appended to this response to make it complete.

We have also tried to address specific comments of Reviewer 02 below. Thanks for the opportunity to revise the manuscript.

**Comments 01:**

1. This manuscript is not up to publishing standards in its current form.

**Response 01:** We have revised the manuscript and ran it also through an editor. Please find appended an updated version of the manuscript.

**Comments 02:**

2. Organization: there is way too many sections and subsection making it hard to follow and understand. Some sections sound repetitive, while other look out of place. There is a number of places in the manuscript that needs serious edits and reorganization.

**Response 02:**

We have tried to improve organization in this version of manuscript (see changes in blue). We have two aspects of data synthesis in a single paper. 1) How to synthesize 1 Hz data from 10-minute wind energy standard data when four statistics, and 2) How to connect two 10-minutes time stamps. We also present a comparison with a similar method to make it complete. This could be one reason why we have many sections.

**Comments 03:**

References in the text are not consistent and should be reformatted.

**Response 03:**

We have updated all the references into an APA style using MS Word template, and now is consistent all across. The typesetter just needs to update style of the reference just at one location to change reference style.

**Comments 04:**

Figures: figures are not up to expected standards. Why some of the plots have a time stamp and why are they differently formatted? Many figures are missing legend and/or labels. There should be better

35    explanation why authors picked to present that specific figure and how it fits in the story they are trying to portrait to a reader.

**Response 04:**

We have updated the following figures/plots.

Figure 1: Completely redrawn and time stamps and labelling issues are addressed.

40    Figure 2: We updated labels and used consistent color codes.

Figure 4: Better plot and distinct color code PSD (Mas) and PSD (Fit) cases.

**Comments 05:**

Equations/math: equations and variables should be properly formatted and explained. Every variable in a
45    equation should be explained and properly formatted, so there is no confusion if it is mentioned in the text (like in line 177 for ais ).

Also, every abbreviations should be explained/defined before used in the text (no matter how trivial they might be to the author, for a reader that might not be a "common knowledge").

**Response 05:**

50    We updated line 177 for ais – subscript/superscript formatting errors.

We reworked on Methodology section and defined als the variable properly. Also, Section 4.1 is updated following reviewers inputs. These changes are marked in blue in the revised manuscript.

**Comments:** Overall, while there is a potential, I feel this should be considered more like a draft version of a manuscript, and less as a final product. I would encourage authors to "build up" from this stage up
55    and spend more time on building and editing pieces that need attention in order to create a manuscript that is up to publication standard.

**Overall Response:** We revised the manuscripts following all the comments and resubmitting a revised version (final product) for the possible publication. Your comments helped improve the quality of the
60    manuscript.

Thank you,

**Ram Poudel,** Corresponding Author

8/12/2023

65